# Modulating Role of Resveratrol in Metabolic and Inflammatory Dysregulation Caused by Surgical and Psychoemotional Stress in Rats

**DOI:** 10.3390/pathophysiology32040067

**Published:** 2025-12-01

**Authors:** Roman Ryabushko, Heorhii Kostenko, Oleh Akimov, Vitalii Kostenko

**Affiliations:** Department of Pathophysiology, Poltava State Medical University, 36011 Poltava, Ukraine; r.riabushko@pdmu.edu.ua (R.R.); heorhiykostenko@gmail.com (H.K.); v.kostenko@pdmu.edu.ua (V.K.)

**Keywords:** resveratrol, single prolonged stress, post-traumatic stress disorder, surgical trauma, systemic inflammatory response, cytokines, insulin resistance, lipid metabolism, antioxidant defense, rats

## Abstract

**Objectives:** This study investigates the effects of resveratrol on systemic inflammatory, oxidative, and metabolic responses in a rat model that combines surgical trauma with prior exposure to Single Prolonged Stress (SPS), an established experimental protocol for modeling post-traumatic stress disorder (PTSD). **Methods:** Male Wistar rats (n = 21) were randomly assigned to three groups: (I) control (polyvinylpyrrolidone, PVP), (II) SPS + laparotomy + PVP), and (III) SPS + laparotomy + resveratrol. Resveratrol (5 mg/kg of body weight/day) or vehicle was administered intragastrically for seven days. Serum concentrations of cortisol, tumor necrosis factor-alpha (TNF-α), interleukin-6 (IL-6), interleukin-10 (IL-10), glucose, insulin, lipid fractions, and thiobarbituric acid–reactive substances (TBA-RS) were determined by enzyme-linked immunosorbent assay and spectrophotometric methods. Insulin resistance was assessed using the homeostatic model assessment of insulin resistance (HOMA-IR) index. **Results:** Combined SPS and surgical trauma induced a pronounced systemic inflammatory response characterized by elevated cortisol (+138%), TNF-α (+83%), IL-6 (+465%), and ceruloplasmin (+71%), as well as hyperglycemia, hyperinsulinemia, increased HOMA-IR, and atherogenic dyslipidemia with reduced high-density lipoprotein cholesterol (HDL-CH; −64%), elevated triglycerides (TGs; +216%), and very low-density lipoprotein cholesterol (VLDL-CH; +218%). Marked activation of lipid peroxidation was observed, as indicated by increased TBA-RS levels before and after incubation. Resveratrol administration significantly decreased cortisol (−45%), TNF-α (−47%), and IL-6 (−85%), normalized the IL-10/IL-6 ratio, and reduced ceruloplasmin levels (−13%). The compound improved insulin sensitivity (HOMA-IR −50%), elevated HDL-CH (+115%), and lowered TGs and VLDL-CH (−44%). It also attenuated both basal and inducible lipid peroxidation (TBA-RS −11% and −13%), indicating restoration of antioxidant capacity. **Conclusions:** Thus, resveratrol effectively counteracts the neuroendocrine, inflammatory, and metabolic disturbances induced by combined PTSD-like stress and surgical trauma.

## 1. Introduction

The ongoing war in Ukraine has dramatically increased the prevalence of psychological trauma among both civilians and military personnel, with a corresponding rise in post-traumatic stress disorder (PTSD) [1,2]. While PTSD is classically defined as a psychiatric disorder, emerging evidence supports its broader classification as a systemic condition characterized by neuroendocrine dysregulation, chronic inflammation, oxidative stress, and metabolic disturbances [3,4,5]. The link between PTSD and systemic inflammation in humans remains complex. While some studies report elevated inflammatory markers in PTSD patients, this association appears to be significant only in certain subgroups [4,5]. These discrepancies necessitate further investigation using preclinical models capable of isolating the contributions of psychological stress to systemic inflammatory responses.

The Single Prolonged Stress (SPS) paradigm is a well-established and widely used rodent model for mimicking PTSD-like states [6]. SPS induces a series of behavioral, hormonal, and immune alterations that align with the core pathophysiological features of PTSD, including hypothalamic–pituitary–adrenal (HPA) axis dysregulation, glucocorticoid receptor desensitization, and heightened pro-inflammatory signaling [7,8]. Rodents subjected to the SPS exhibit impaired retention of extinction memory, mirroring a core characteristic observed in individuals with PTSD, wherein fear responses persist despite successful extinction training [9]. Despite occasional concerns regarding its translatability—given the absence of formal diagnostic criteria for PTSD in animals—SPS remains a robust experimental approach, validated across multiple studies involving pharmacological and molecular endpoints [10].

Surgical trauma—often unavoidable in conflict-related injuries—can further exacerbate the systemic inflammatory response (SIR), particularly when occurring alongside pre-existing psychological stress. This combination accelerates the overproduction of reactive oxygen and nitrogen species (ROS/RNS), disrupts redox homeostasis, and impairs mitochondrial function [11,12]. Collectively, these alterations contribute to the development of a systemic inflammatory response phenotype, characterized by sustained elevation of pro-inflammatory cytokines, acute-phase proteins, and connective tissue damage [13,14].

Redox-sensitive transcription factors such as nuclear factor kappa B (NF-κB), nuclear factor erythroid 2-related factor 2 (Nrf2), signal transducer and activator of transcription 3 (STAT3), and activator protein 1 (AP-1) play central roles in regulating stress- and inflammation-related signaling pathways implicated in PTSD pathogenesis [15,16,17,18]. Their dysregulation perpetuates a “vicious cycle” of inflammation, facilitating the progression of chronic disorders including neurodegeneration, metabolic syndrome, cardiovascular disease, and persistent intestinal inflammation [14].

Although advances in anti-inflammatory therapies have been made, most current treatments target downstream effectors and fail to modulate upstream transcriptional regulators. Additionally, synthetic modulators of transcription factors are often associated with adverse side effects. In this context, natural polyphenols such as resveratrol offer promising therapeutic potential due to their multi-target activity, favorable safety profile, and ability to concurrently modulate several pro-inflammatory transcription factors [14,19,20].

Resveratrol, a stilbene-class polyphenol, is particularly notable for its ability to activate Sirtuin 1 (Sirt1)—a NAD^+^-dependent protein deacetylase that regulates a wide range of transcriptional networks involved in oxidative stress and inflammation [21]. Although a recent systematic review and dose–response meta-analysis of randomized controlled trials in humans did not find a consistent overall effect of resveratrol on Sirt1 expression or serum levels, subgroup analyses indicated significant upregulation in studies with shorter intervention durations and specific tissue targets, suggesting that the effect may be dose- and context-dependent [22]. Through Sirt1-dependent mechanisms, resveratrol influences key regulatory proteins including STAT3, Forkhead box O1 and O3 (FOXO1/3), tumor protein p53, hairy/enhancer-of-split related to YRPW motif protein 2 (HEY2), and peroxisome proliferator-activated receptor gamma (PPARγ), thereby attenuating inflammatory signaling and restoring redox balance [23]. Experimental data further support the ability of resveratrol to reduce nitro-oxidative stress and downregulate pro-inflammatory cytokines, particularly those driven by tumor necrosis factor-alpha (TNF-α) [24].

Resveratrol therapeutic efficacy in various inflammatory and metabolic conditions such as myocardial ischemia, type 2 diabetes, metabolic syndrome, and non-alcoholic steatohepatitis has been extensively documented [25,26]. However, many of the upstream molecular mechanisms remain to be fully elucidated [27]. These findings suggest that resveratrol protective actions are mediated through a combination of anti-inflammatory, antioxidant, and metabolic regulatory pathways.

Despite promising preclinical data, the therapeutic translation of resveratrol faces significant limitations. It exhibits extremely low oral bioavailability (often less than 1% of the administered dose), rapid metabolism, poor solubility, chemical instability, and potential adverse effects, all of which constrain its clinical applicability [28]. These pharmacokinetic challenges likely contribute to the controversial and inconsistent outcomes reported in clinical trials, where resveratrol supplementation has shown variable or negligible effects on key biomarkers. This highlights the need for more rigorous, well-powered studies employing optimized dosing regimens and improved delivery systems [29].

While the anti-inflammatory and metabolic effects of resveratrol have been documented in various chronic disease models, its role in complex stress-driven conditions that combine prior psychological stress with subsequent surgical injury remains underexplored. Existing studies have largely focused on isolated forms of stress, neglecting the potentially synergistic burden imposed by their co-occurrence. Moreover, the SPS model—though primarily behavioral—is increasingly recognized for its capacity to induce persistent systemic inflammatory changes distinct from acute stress exposure. To date, no study has comprehensively examined whether resveratrol can attenuate the compounded systemic disturbances arising from this dual-stressor model. Addressing this gap may provide novel insights into resveratrol’s relevance for stress-associated somatic pathologies.

Therefore, the aim of this study is to assess the effects of resveratrol on systemic inflammatory, oxidative, and metabolic responses in a rat model combining surgical trauma with prior stress exposure via the SPS protocol. While SPS is widely recognized as a preclinical model for PTSD-like states with distinct pro-inflammatory potential, the primary focus of this work is on somatic stress parameters relevant to stress-associated comorbidities

## 2. Materials and Methods

### 2.1. Animals and Ethics Statement

The study was conducted on 21 adult male Wistar rats aged 8–10 weeks and weighing 210–230 g at the beginning of the experiment. Animals were housed in polycarbonate cages (two animals per cage) under standard vivarium conditions: ambient temperature of 22 ± 2 °C, relative humidity of 30–60%, and a 12:12 h light/dark cycle (lights on at 7:00 a.m.). Rats had free access to standard laboratory chow and tap water ad libitum. Cage bedding was replaced twice weekly, and environmental enrichment (paper rolls, nesting material) was provided to minimize stress.

To reduce animal distress, all procedures were conducted during the light phase in a quiet room by experienced personnel. Animals were acclimatized for 7 days before the start of the experiment, monitored daily for signs of discomfort, and handled regularly to reduce handling-related stress. During experimental manipulations, care was taken to use the least invasive techniques consistent with scientific objectives. Anesthesia and euthanasia were performed in accordance with institutional and international ethical standards to minimize pain and distress. All experimental procedures were carried out in compliance with the European Convention for the Protection of Vertebrate Animals Used for Experimental and Other Scientific Purposes (Strasbourg, 1986), the provisions of EU Directive 2010/63/EU, and relevant national legislation. The study protocol was reviewed and approved by the Commission on Bioethics and Ethical Issues of Poltava State Medical University (Protocol No. 213, 22 February 2023).

### 2.2. Experimental Design

Rats were randomly divided into three experimental groups (n = 7 per group). Group I (Control, intact animals) received polyvinylpyrrolidone (PVP) at a dose of 180 mg/kg of body weight (bw), administered once daily by intragastric gavage for seven consecutive days. The PVP solution served as a vehicle and solubility enhancer for stilbenes, and its dosing regimen was based on previously published protocols [30].

Animals in Groups II and III were subjected to the SPS followed by laparotomy. Seven days after SPS exposure, all rats underwent behavioral screening using the open field test to evaluate anxiety-like responses. Animals that demonstrated characteristic PTSD-like behaviors—including reduced time in the central zone (less than 20% of total test time), increased thigmotaxis, decreased locomotor activity, and prolonged freezing—were considered to meet inclusion criteria for PTSD-like status [31,32]. Only rats fulfilling these criteria were randomized into Group II (SPS + laparotomy + vehicle) and Group III (SPS + laparotomy + resveratrol), thereby ensuring behavioral homogeneity across stress-exposed groups.

Starting 24 h after surgery, these animals received daily intragastric administration of the test formulations for seven consecutive days. Group II (received the vehicle solution (PVP, 180 mg/kg bw in 0.25% HPMC), while Group III received resveratrol at a dose of 5 mg/kg bw [33], also formulated with PVP (180 mg/kg bw) in 0.25% HPMC. All treatments were administered once daily via oral gavage at a dosing volume of 5 mL/kg. Randomization and outcome assessments were conducted by investigators blinded to group allocation to minimize experimental bias.

Resveratrol (≥98% purity, Shaanxi Jiahe Phytochem Co., Fengdong new town, Xixian New District, Xi’an, China) and polyvinylpyrrolidone K30 (PVP K30, Yuking Technologies Co., Shanghai, China) were used for formulation. Aqueous suspensions were freshly prepared each day of dosing. PVP K30 was dissolved in purified water containing 0.25% (*w*/*v*) hydroxypropyl methylcellulose (HPMC, Wanhua Chemical Co., Jinan, China) to yield a final concentration of 36 mg/mL. Resveratrol was then incorporated to achieve a final concentration of 1 mg/mL. The suspension was gently stirred until homogeneous and protected from light throughout preparation and administration. Animals received the formulation by intragastric gavage at 5 mg/kg resveratrol and 180 mg/kg PVP in a dosing volume of 5 mL/kg. The vehicle control consisted of the corresponding PVP solution (36 mg/mL in 0.25% HPMC) without resveratrol, administered under identical conditions.

After seven days of treatment, animals were anesthetized with thiopental sodium and euthanized in accordance with institutional ethical standards. Blood samples were collected immediately by cardiac puncture into sterile tubes without anticoagulant. Samples were allowed to clot at room temperature for 30 min and then centrifuged at 3000× *g* for 15 min. The resulting serum was carefully separated and stored for subsequent biochemical analyses.

### 2.3. Experimental Models

To induce SPS, rats were immobilized for 2 h on a metal platform with their limbs secured using surgical tape. Following immobilization, the animals were subjected to a forced swim test in a plexiglas cylinder filled to two-thirds with fresh water maintained at 24 °C. Subsequently, the rats were exposed to sevoflurane vapor (Sevoran, AbbVie S.r.l., Campoverde di Aprilia, Italy) until loss of consciousness. After the procedure, animals were housed in pairs and left undisturbed for 7 days [7]. According to current literature, this SPS protocol reliably induces behavioral and functional changes consistent with PTSD [8].

Laparotomy was performed as previously described [12] under intraperitoneal anesthesia with sodium thiopental (Kyivmedpreparat, Kyiv, Ukraine) at a dose of 50 mg/kg bw. After antiseptic preparation of the abdominal skin, a 1 cm midline incision was made in the lower abdomen, followed by blunt dissection through the muscle layers, fascia, and peritoneum. A loop of the small intestine was exteriorized and gently massaged for 10 s before being repositioned into the abdominal cavity. The incision was closed in layers using polyglycolide suture material with an atraumatic needle (Biopolymer, Poltava, Ukraine).

### 2.4. Biochemical and Enzyme-Linked Immunosorbent Assays

Serum cortisol levels (nmol/L), used as a marker of acute stress response, were determined spectrophotometrically using a ULAB 101 spectrophotometer (ULAB, Nanjing, China). The assay was based on the reaction of cortisol with nitroblue tetrazolium (IUPAC name: 2,2′-bis(4-Nitrophenyl)-5,5′-diphenyl-3,3′-(3,3′-dimethoxy-4,4′-diphenylene) ditetrazolium chloride, Sigma Aldrich, St. Louis, MO, USA) in methanol, in the presence of tetramethylammonium hydroxide pentahydrate, resulting in the formation of a red–orange chromophore with a maximum absorbance at 510 nm, as previously described [34]. Serum cytokine levels were measured using highly sensitive and rat-specific enzyme-linked immunosorbent assay (ELISA) kits: Rat TNF-α ELISA Kit (RAB0479-1KT), Rat IL-6 ELISA Kit (RAB0311-1KT), and Rat IL-10 ELISA Kit (RAB0246-1KT), all from Sigma-Aldrich (USA). Absorbance was recorded at 450 nm using a Stat Fax 2100 microplate reader (Awareness Technology, Inc., Palm City, Martin, FL, USA), following the manufacturer’s instructions. Serum ceruloplasmin levels, a marker of the acute-phase inflammatory response, were measured using a colorimetric method based on the oxidation of p-phenylenediamine, as previously described [35].

Serum glucose concentrations were determined using the glucose oxidase method, based on the formation of a colored quinonimine compound. The absorbance was measured spectrophotometrically at 540 nm using a commercial assay kit (HP009.02) from Filicit-Diagnostics (Dnipro, Ukraine). Fasting serum insulin levels were quantified using a Rat Ins1/Insulin ELISA Kit (RAB0904-1KT, Sigma-Aldrich, USA) according to the manufacturer’s protocol. Blood samples were collected after a 12 h overnight fast. Absorbance was measured at 450 nm with a Stat Fax 2100 microplate reader (Awareness Technology, Inc., USA). Insulin resistance was calculated using the homeostatic model assessment of insulin resistance (HOMA-IR) index, according to the equation: HOMA-IR = (Fasting glucose (mmol/L) × Fasting insulin (μU/mL))/22.5 [36].

Total cholesterol (CH, TC0102), high-density lipoprotein cholesterol (HDL, HDL0102), and triacylglycerols (TGs, TG0102) were measured using enzymatic colorimetric methods with standardized reagent kits (Mindray Bio-Medical Electronics Co., Nanshan, Shenzhen, China). Absorbance was recorded using a ULAB 101 spectrophotometer (ULAB, Nanjing, China), capable of measuring in the 490–600 nm wavelength range. Low-density lipoprotein (LDL) and very low-density lipoprotein (VLDL) concentrations were calculated using the Friedewald equation.

Lipid peroxidation (LPO) was evaluated by measuring thiobarbituric acid reactive substances (TBA-RS) levels, determined through the formation of malondialdehyde-thiobarbituric acid adducts, with peak absorbance at 532 nm. Measurements were performed using a ULAB 101 spectrophotometer (ULAB, Nanjing, China). The overall antioxidant capacity of blood plasma was estimated based on the increase in TBA-RS concentration after 90 min incubation in a pro-oxidant ascorbate–iron buffer (pH 7.4; composition per liter: 1.9 g Tris-HCl, 50 mL of 0.1 N HCl, 1.4 g ascorbic acid, and 32 mg ferrous sulfate heptahydrate) [37].

### 2.5. Statistical Analysis

All statistical analyses were performed using Microsoft Excel (Microsoft 365) with the Real Statistics Resource Pack add-in (version 2019). Prior to conducting parametric tests, the assumption of normality was evaluated for each dataset using the Shapiro–Wilk test, which is particularly suitable for small to moderate sample sizes and is widely recommended for assessing data normality in bio-medical research. The test was applied to each experimental group separately, and visual inspection of histograms and Q-Q plots was also conducted to support the interpretation of normality. Only datasets that passed the Shapiro–Wilk test (*p* > 0.05) were subjected to parametric statistical procedures. Results are presented as mean ± standard error of the mean (SEM). For data conforming to normal distribution, one-way analysis of variance (ANOVA) was used, followed by post hoc pairwise comparisons with Student’s *t*-test for independent samples and Tukey’s honestly significant difference (HSD) test. The Dunn–Šidák correction was applied to adjust for multiple comparisons and control the family-wise error rate. Statistical significance was defined as *p* < 0.05.

## 3. Results

### 3.1. Effect of Resveratrol on the Indicators of Acute Stress and Systemic Inflammatory Response in Blood Serum Under Experimental Surgical Trauma and Post-Traumatic Stress Disorder

Exposure to the combined effects of SPS and subsequent laparotomy caused a pronounced activation of neuroendocrine and inflammatory pathways, as evidenced by significant changes in serum stress and immune markers (Table 1). In the SPS + laparotomy group (Group II), serum cortisol concentrations increased by 138% compared with intact control animals (*p* < 0.001), indicating marked hyperactivation of the hypothalamic–pituitary–adrenal (HPA) axis. Resveratrol administration (Group III) reduced cortisol levels by 44.9% relative to Group II (*p* < 0.001), restoring values close to those of the control group.

Serum TNF-α increased by 82.6% and IL-6 by 465% in the SPS + laparotomy rats versus controls (both *p* < 0.001), confirming the induction of a strong SIR. Treatment with resveratrol reduced TNF-α levels by 46.7% and IL-6 levels by 84.5% relative to untreated stressed animals (*p* < 0.001). These changes clearly demonstrate potent ability of resveratrol to suppress pro-inflammatory cytokine production under conditions of combined psychological and surgical stress.

The anti-inflammatory cytokine interleukin-10 (IL-10) showed a moderate compensatory rise of 48.8% in the SPS + laparotomy group compared with controls (*p* < 0.01). However, following resveratrol treatment, IL-10 levels decreased by 45.2% relative to Group II (*p* < 0.001), resulting in a more balanced IL-10/IL-6 ratio indicative of restored cytokine homeostasis.

The IL-10/IL-6 ratio, a marker of the balance between anti- and pro-inflammatory cytokines, was reduced by 72.4% in rats subjected to SPS and laparotomy compared to control animals (*p* < 0.01), indicating a pronounced shift toward a pro-inflammatory state. Administration of resveratrol restored this ratio to near-control levels and exceeded the value observed in Group II by 237.5% (*p* < 0.001), suggesting a potent anti-inflammatory effect.

Serum ceruloplasmin, an acute-phase protein and marker of oxidative stress, increased by 71.1% after SPS combined with surgery (*p* < 0.001). Resveratrol treatment reduced ceruloplasmin levels by 13% compared with Group II (*p* < 0.001), reflecting attenuation of the acute-phase response and normalization of redox balance.

Overall, these findings demonstrate that the combination of PTSD-like stress and surgical trauma induces a synergistic neuroinflammatory response characterized by endocrine hyperactivation, cytokine overproduction, and redox imbalance. Resveratrol administration markedly counteracted these effects, reducing stress hormone levels, suppressing pro-inflammatory cytokines, and restoring antioxidant homeostasis in blood serum.

### 3.2. Effect of Resveratrol on Carbohydrate Metabolism Parameters in the Blood Serum Under Experimental Surgical Trauma and Post-Traumatic Stress Disorder

Alterations in carbohydrate metabolism were observed in rats exposed to SPS followed by surgical trauma (Table 2). The combination of psychological and physical stressors induced marked disturbances in glucose–insulin homeostasis consistent with insulin resistance and metabolic stress. In the SPS + laparotomy group (Group II), serum glucose concentrations increased by 34% compared with the control group (*p* < 0.001), indicating stress-induced hyperglycemia. Administration of resveratrol resulted in a 24.4% reduction in blood glucose levels compared to Group II (*p* < 0.05).

A similar trend was observed for serum insulin, which exhibited a dramatic 329% increase after SPS and surgical trauma compared with controls (*p* < 0.01). This hyperinsulinemia reflected compensatory overactivation of pancreatic β-cells in response to peripheral insulin resistance. Treatment with resveratrol did not produce a statistically significant change in insulin levels compared with the SPS + laparotomy group; the observed differences were not significant, confirming the null hypothesis (*p* > 0.05).

Consistent with these findings, the HOMA-IR index, a standard indicator of insulin resistance, increased nearly 5.7-fold under combined SPS and surgical trauma compared with intact controls (*p* < 0.01). Resveratrol treatment reduced HOMA-IR by 50% relative to Group II (*p* < 0.05), confirming a substantial improvement in insulin sensitivity and normalization of carbohydrate metabolism.

Overall, these data demonstrate that the coexistence of post-traumatic stress and surgical injury leads to pronounced metabolic derangements characterized by hyperglycemia, hyperinsulinemia, and elevated insulin resistance. Resveratrol markedly alleviated these metabolic disturbances, indicating its potential to modulate stress-associated metabolic pathways through mechanisms involving improved insulin signaling.

### 3.3. Effect of Resveratrol on Lipid Profile in the Blood Serum Under Experimental Surgical Trauma and Post-Traumatic Stress Disorder

Significant alterations in lipid metabolism were observed in rats exposed to SPS followed by surgical trauma (Table 3). These changes reflected the development of a stress-induced dyslipidemic profile characterized by hypertriglyceridemia, accumulation of VLDL-CH, and depletion of HDL-CH.

In the SPS + laparotomy group (Group II), serum CH values did not differ significantly from those in the control group (*p* > 0.05). However, the anti-atherogenic fraction HDL-CH decreased sharply, by about 64.4% relative to control values (*p* < 0.001). At the same time, VLDL-CH and TGs rose markedly, by 218% and 216%, respectively (both *p* < 0.001). No statistically significant differences in LDL-CH levels were observed between the SPS + laparotomy group and the control group, confirming the null hypothesis (*p* > 0.05).

Treatment with resveratrol effectively ameliorated these lipid abnormalities. Administration of the compound increased HDL-CH levels by more than 115% compared with untreated SPS + laparotomy rats (*p* < 0.001), partially restoring values toward the control range. Concurrently, VLDL-CH and TGs were both reduced by 44.2% relative to Group II (*p* < 0.001). Total and LDL-CH levels did not differ significantly between the resveratrol-treated and SPS + laparotomy groups (*p* > 0.05).

Thus, these results demonstrate that combined psychological and surgical stress induces a pronounced dyslipidemic shift characterized by increased triglyceride and VLDL-CH levels and depleted HDL-CH. Resveratrol administration significantly counteracted these alterations, improving the lipid profile primarily through normalization of HDL-CH and VLDL-CH fractions and attenuation of stress-related hypertriglyceridemia. These findings highlight resveratrol ability to restore lipid homeostasis under conditions of SIR and metabolic stress.

### 3.4. Effect of Resveratrol on the Secondary Lipid Peroxidation Products in the Blood Under Experimental Surgical Trauma and Post-Traumatic Stress Disorder

Indicators of lipid peroxidation were markedly altered in rats subjected to SPS followed by surgical trauma (Table 4). The concentration of thiobarbituric acid-reactive substances (TBA-RS), which reflect the accumulation of secondary lipid peroxidation products such as malondialdehyde, demonstrated a significant increase both before and after incubation in pro-oxidant ascorbate–iron buffer, indicating pronounced activation of oxidative stress processes in the blood.

In the SPS + laparotomy group (Group II), the baseline (pre-incubation) level of TBA-reactive compounds increased by approximately 56.9% compared with control animals (*p* < 0.001). After incubation in a pro-oxidant medium, the concentration rose further, showing a 59% elevation over the control group (*p* < 0.001). The increment over incubation time, which represents the susceptibility of plasma lipids to oxidative damage, also increased significantly, by 62% relative to controls (*p* < 0.01). These results indicate that combined psychological and surgical stress induces both enhanced lipid peroxidation and decreased antioxidant stability of circulating lipoproteins.

Treatment with resveratrol effectively reduced oxidative lipid degradation. In resveratrol-treated rats (Group III), the pre-incubation TBA-RS level was 11% lower than in the SPS + laparotomy group (*p* < 0.001), while the post-incubation value decreased by 12.9% (*p* < 0.001). The incubation-dependent increment was also reduced by approximately 16% compared with Group II (*p* < 0.05). These findings suggest that resveratrol mitigates both the basal and inducible components of lipid peroxidation, indicating an enhancement of systemic antioxidant defenses and stabilization of cell membrane lipids.

The data obtained demonstrate that combined post-traumatic stress and surgical injury significantly increase oxidative modification of plasma lipids, whereas resveratrol administration attenuates this process, reflecting its capacity to restore redox balance and limit free radical-mediated damage under SIR conditions.

## 4. Discussion

The SPS paradigm remains one of the most widely accepted and extensively validated rodent models for inducing PTSD-like states [6,10]. While no preclinical model can fully replicate the complexity of human PTSD, SPS reliably induces core pathophysiological hallmarks of the disorder, including HPA axis dysregulation, glucocorticoid receptor desensitization, heightened pro-inflammatory signaling, and behavioral alterations such as persistent anxiety-like responses [7,8]. In the present study, rigorous behavioral screening following SPS exposure ensured the inclusion of animals exhibiting well-defined PTSD-like phenotypes, as determined by open field test criteria previously validated for modeling anxiety and hypervigilance in this context.

The superimposition of surgical trauma on this PTSD-like background offers a clinically relevant scenario that mimics comorbid somatic injury in psychologically stressed individuals—a common occurrence in both civilian and military populations. This model enabled the evaluation of systemic inflammatory, oxidative, and metabolic disruptions under conditions that closely reflect the interplay between psychological and physical stressors. Investigating these perturbations is essential for understanding how pre-existing psychological stress modulates the host response to trauma, potentially exacerbating systemic inflammation and metabolic dysfunction. Moreover, this experimental framework provides a valuable platform for testing therapeutic agents, such as resveratrol, that may attenuate stress- and injury-induced pathophysiology through pleiotropic anti-inflammatory and antioxidant mechanisms.

This study demonstrates that resveratrol exerts a pronounced protective effect against SIR, metabolic dysregulation, and oxidative stress induced by the combination of PTSD-like conditions and surgical trauma. The obtained results suggest that the coexistence of psychological stress and tissue injury contributes to the activation of the HPA axis, as reflected by elevated cortisol levels, and is associated with cytokine imbalance and metabolic disturbances.

The observed elevation of serum TNF-α and IL-6 in the SPS + laparotomy group is consistent with previous findings demonstrating that chronic restraint stress over 7 to 14 days promotes a sustained pro-inflammatory state, as evidenced by progressive increases in plasma IL-6, TNF-α, C-reactive protein levels, and hepatic IL-6 expression [38]. The authors concluded that integrating this stress paradigm with trauma or sepsis models may provide valuable insights into the role of persistent inflammation in disease progression and clinical outcomes. PTSD is known to disrupt the normal negative feedback of cortisol on cytokine production, thereby amplifying the inflammatory cascade [39]. Moreover, surgery itself is a potent trigger of SIR, mediated by tissue damage, mitochondrial dysfunction, and activation of innate immune pathways [12].

Resveratrol significantly reduced cortisol, TNF-α, and IL-6 levels, suggesting modulation of both HPA axis activity and cytokine release. Similar anti-inflammatory effects of resveratrol have been documented in SIR models and metabolic syndrome, where the compound attenuated NF-κB activation and increased nuclear translocation of Nrf2 [40,41,42]. The observed normalization of the IL-10/IL-6 ratio in our study further supports a shift toward anti-inflammatory homeostasis, consistent with the regulatory actions of Sirt1-dependent deacetylation of NF-κB p65 subunits [43].

The SPS + laparotomy model produced marked hyperglycemia, hyperinsulinemia, and increased HOMA-IR, reflecting insulin resistance secondary to stress-induced cytokine release and lipid peroxidation. Activation of the NF-κB/IκB kinase complex signaling pathway, potentially induced by TNF-α and IL-6, leads to serine phosphorylation of insulin receptor substrate 1 (IRS-1), which disrupts normal insulin signaling (e.g., by causing dissociation of IRS-1 from the insulin receptor) and contributes to the development of insulin resistance [44].

Although resveratrol did not significantly affect fasting glucose or insulin levels, the reduction of HOMA-IR indicates a partial restoration of insulin sensitivity. These findings are in agreement with reports showing that resveratrol enhances insulin signaling by activating AMP-activated protein kinase and Sirt1, leading to improved glucose utilization and mitochondrial function [45].

The combined exposure to PTSD-like stress and surgery induced a distinct dyslipidemic profile characterized by elevated TGs and VLDL-CH and markedly reduced HDL-CH. This lipid pattern reflects enhanced hepatic lipogenesis and impaired reverse cholesterol transport, typical for chronic systemic inflammation [46].

Resveratrol administration normalized HDL-CH levels and significantly reduced TGs and VLDL-CH, consistent with its known ability to stimulate PPARα and liver X receptor alpha (LXR-α) pathways and to enhance apolipoprotein A-I synthesis [47]. These effects improve HDL function and promote cholesterol efflux from macrophages, thus exerting anti-atherogenic and anti-inflammatory actions [48]. The absence of significant changes in total and LDL-CH concentrations suggests that resveratrol primarily affects triglyceride-rich fractions and HDL metabolism, rather than total cholesterol turnover, in acute stress settings.

The substantial rise in TBA-reactive substances observed in SPS + laparotomy animals confirms intense lipid peroxidation and redox imbalance, consistent with our previous findings where elevated reactive oxygen species (ROS) and nitric oxide-derived radicals contributed to cellular and mitochondrial injury [49]. The increased post-incubation increment of TBA-RS further indicates reduced antioxidant reserve and heightened susceptibility of plasma lipids to oxidative damage.

Resveratrol treatment significantly reduced both basal and inducible lipid peroxidation. This observation supports earlier reports that resveratrol acts as a chain-breaking antioxidant and enhances endogenous defense systems through Nrf2 activation, leading to increased expression of heme oxygenase-1, superoxide dismutase, and glutathione peroxidase [50]. Beyond direct free-radical scavenging, resveratrol may stabilize cell membranes by modulating lipid bilayer fluidity and inhibiting NADPH oxidase-dependent superoxide production [51].

The cumulative data from our study support a multifaceted protective role of resveratrol in the context of combined psychological and surgical stress. This effect may be attributed to the ability of the compound to act through multiple molecular targets, including suppression of NF-κB-mediated pro-inflammatory cytokine transcription, activation of Sirt1/AMP activated protein kinase and Nrf2-dependent antioxidant pathways, and modulation of lipid and glucose metabolism [52,53]. These coordinated effects restore systemic homeostasis, limiting both inflammatory and metabolic damage. The observed normalization of lipid peroxidation and cytokine profiles highlights potential of resveratrol as a pleiotropic modulator of the systemic inflammatory response, especially relevant for stress-related and trauma-induced disorders.

*Study limitations*. This study has several limitations. The absence of a surgical-only control group without prior SPS exposure restricts direct differentiation between the individual contributions of psychological and surgical stress. Nevertheless, the current dataset provides robust biochemical evidence supporting integrative anti-inflammatory and metabolic-stabilizing actions of resveratrol in this complex comorbid model. The small sample size and use of only male animals may limit the generalizability of the findings and prevent detection of sex-specific responses. In addition, the short duration of follow-up does not allow conclusions about the long-term efficacy of resveratrol in this model.

## 5. Conclusions

Thus, the combined surgical trauma and PTSD experimental model (SPS) in rats induces a pronounced systemic inflammatory and metabolic response, characterized by hyperactivation of the HPA axis, elevated pro-inflammatory cytokines (TNF-α, IL-6), oxidative stress, and lipid peroxidation, reflecting the development of a systemic inflammatory response phenotype.

This comorbid model produces marked metabolic dysfunction, including hyperglycemia, hyperinsulinemia, increased insulin resistance, and dyslipidemia, with decreased HDL-CH and elevated TGs and VLDL-CH, accompanied by impaired redox balance.

Resveratrol administration effectively mitigates these disturbances by attenuating systemic inflammation, lowering cortisol as well as pro-inflammatory cytokines (TNF-α and IL-6), enhancing IL-10 as a pivotal anti-inflammatory mediator, improving antioxidant defenses, and restoring key indicators of metabolic stress.

The study demonstrates that resveratrol acts as a potent regulator of redox and metabolic homeostasis under conditions of combined psychological and surgical trauma, suggesting its therapeutic potential as an adjunctive agent for preventing systemic inflammatory and metabolic complications associated with PTSD and surgical injury.

## Figures and Tables

**Table 1 pathophysiology-32-00067-t001:** Effect of resveratrol on the indicators of acute stress and systemic inflammatory response in serum under experimental surgical trauma and post-traumatic stress disorder.

Parameters	Control Rats (only PVP), n = 7	SPS + Subsequent Laparotomy	
		+PVP, n = 7	+Resveratrol + PVP, n = 7
Cortisol (nmol/L)	15.39 ± 0.67	36.71 ± 0.60 *	20.23 ± 0.31 *^,^**
TNF-α (pg/mL)	235.8 ± 7.9	430.6 ± 26.1 *	229.7 ± 12.7 **
IL-6 (pg/mL)	79.3 ± 10.3	448.1 ± 62.3 *	69.4 ± 11.1 **
IL-10/IL-6 ratio	2.9 ± 0.5	0.8 ± 0.1 *	2.7 ± 0.3 **
IL-10 (pg/mL)	203.8 ± 18	303.3 ± 22.4 *	166.2 ± 7.7 **
Ceruloplasmin (mg/L)	308.4 ± 6.5	527.6 ± 3.1 *	458.8 ± 11.3 *^,^**

Note: The table represents the mean ± SEM; * *p* < 0.05 compared to findings in the control group; ** *p* < 0.05 compared to findings in the rats of Group 2; IL—Interleukin, PVP—Polyvinylpyrrolidone, SPS—Single Prolonged Stress, TNF-α—Tumor Necrosis Factor-alpha.

**Table 2 pathophysiology-32-00067-t002:** Effect of resveratrol on carbohydrate metabolism parameters in the serum under experimental surgical trauma and post-traumatic stress disorder.

Parameters	Control Rats (only PVP), n = 7	SPS + Subsequent Laparotomy	
		+PVP, n = 7	+Resveratrol + PVP, n = 7
Glucose (mmol/L)	4.83 ± 0.26	6.47 ± 0.06 *	4.89 ± 0.20 **
Insulin (μU/mL)	37.1 ± 2.7	159.2 ± 33.1 *	104.9 ± 14.7
Homeostatic Model Assessment of Insulin Resistance (HOMA-IR)	8.0 ± 0.8	45.5 ± 9.3	22.6 ± 3.1 *^,^**

Note: The table represents the mean ± SEM; * *p* < 0.05 compared to findings in the control group; ** *p* < 0.05 compared to findings in the rats of Group 2; PVP—Polyvinylpyrrolidone, SPS—Single Prolonged Stress.

**Table 3 pathophysiology-32-00067-t003:** Effect of resveratrol on lipid profile in the blood serum under experimental surgical trauma and post-traumatic stress disorder.

Parameters	Control Rats (only PVP), n = 7	SPS + Subsequent Laparotomy	
		+PVP, n = 7	+Resveratrol + PVP, n = 7
Total CH (mmol/L)	2.39 ± 0.22	2.73 ± 0.09	2.62 ± 0.23
HDL-CH (mmol/L)	0.90 ± 0.06	0.32 ± 0.03 *	0.69 ± 0.04 *^,^**
LDL-CH (mmol/L)	1.22 ± 0.24	1.55 ± 0.09	1.45 ± 0.23
VLDL-CH (mmol/L)	0.27 ± 0.02	0.86 ± 0.04 *	0.48 ± 0.02 *^,^**
TGs (mmol/L)	0.60 ± 0.05	1.90 ± 0.08 *	1.06 ± 0.04 *^,^**

Note: The table represents the mean ± SEM; * *p* < 0.05 compared to findings in the control group; ** *p* < 0.05 compared to findings in the rats of Group 2; CH—Cholesterol, HDL—High-Density Lipoprotein, LDL—Low-Density Lipoprotein, PVP—Polyvinylpyrrolidone, SPS—Single Prolonged Stress, TGs—Triacylglycerols (Triglycerides), VLDL—Very-Low-Density Lipoprotein.

**Table 4 pathophysiology-32-00067-t004:** Effect of resveratrol on the secondary lipid peroxidation products in the blood under experimental surgical trauma and post-traumatic stress disorder.

Parameters	Control Rats (Only PVP), n = 7	SPS + Subsequent Laparotomy	
		+PVP, n = 7	+Resveratrol + PVP, n = 7
TBA-reacting compounds (µmol/kg):			
Before incubation	11.6 ± 0.2	18.2 ± 0.3 *	16.2 ± 0.2 *^,^**
After incubation	16.6 ± 0.5	26.4 ± 0.5 *	23.0 ± 0.2 *^,^**
Increment over incubation time	5.0 ± 0.7	8.1 ± 0.5 *	6.8 ± 0.3 *^,^**

Note: The table represents the mean ± SEM; * *p* < 0.05 compared to findings in the control group; ** *p* < 0.05 compared to findings in the rats of Group 2; PVP—Polyvinylpyrrolidone, SPS—Single Prolonged Stress, TBA—Thiobarbituric Acid.

## Data Availability

The datasets used and/or analyzed during the current study are available from the corresponding author on reasonable request.

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
