# Peer review of "Modulating Role of Resveratrol in Metabolic and Inflammatory Dysregulation Caused by Surgical and Psychoemotional Stress in Rats"

_pathophysiology, 2025, doi:10.3390/pathophysiology32040067_

Round 1
Reviewer 1 Report
Comments and Suggestions for Authors
In the manuscript entitled “Resveratrol Alleviates Metabolic Disturbances and Systemic 2 Inflammatory Response in Rats with Experimental Surgical 3 Trauma and Post-Traumatic Stress Disorder” the authors, Ryabushko R. et al. present the studies of the effect of resveratrol supplementation on the levels of indicators of acute stress and systemic inflammatory response (cortisol, TNF-α, IL-6, IL-10 and ceruloplasmin) in rats undergoing single prolonged stress and subsequent laparotomy. In addition, the levels of glucose and insulin, and lipid profile in blood serum were determined indicating alleviating effects of resveratrol on the processes triggered by post-traumatic stress disorder (PTSD).
Some corrections would be desirable to make the experimental investigations unambiguous and better described.
My comments:
- Please, clarify the PTSD as a syndrome induced by combined treatment: single pronounced stress (SPS) and laparotomy. The first sentence in the abstract is not clear; one could think that surgical trauma did not induce PTSD.
Line 25; (after incubation) add: in pro-oxidant ascorbate-iron buffer.
- Introduction: it would be worth mentioning the limitations of the use of resveratrol as a therapeutic agent (Ren Z.Q. et al. MedComm (2020). 2025 Jun 11;6(6):e70252), and the controversial results of its biological activities, especially in clinical trials (Drago L. et al. Food Frontiers. 2024;5:849–854) . Mansouri et al. cited in the manuscript [15] concluded that the dose-response meta-analysis of impact of resveratrol supplementation on sirtuin 1 level was not statistically significant.
- 2. Experimental design: line126 and 127 add "body weight" (5 mg/kg body weight).
Line 116; (Group I and II) or (Control group, Group II)
- 4. Biochemical and Enzyme-linked immunosorbent assays.
Melatonin was listed as the study compound (that in fact were not analysed), while, interleukin-10, ceruloplasmin and cortisol were omitted and the assays were not described.
Description of all the methods of biochemical determinations used in the studies, is very important.
The methods of determining of glucose, cholesterol, HDL and TG should be given in more details.
- Results:
Table 2. Are the numbers concerning glucose concentration correct because the difference between 6.47 and 4.89 seems to be significant.
Line 295: precise what incubation you mean (in pro-oxidant ascorbate-iron buffer)?
- Line 345: The ratio IL-10/IL-6 (not IL-6/IL-10) is usually used to assess anti-inflammatory homeostasis. It would be advisable to include this parameter in the table 1.
- Discussion: Paragraph lines 385-393. Sirt 1/AMPK and Nrf2 antioxidant pathways were not investigated in this research, so please, indicate that these information comes from literature giving an appropriate citation.
- Conclusions: line 409; IL-6 is pro-inflammatory cytokine, while IL-10 is anti-inflammatory cytokine (human cytokine synthesis inhibitory factor).
Author Response
We sincerely thank you for your thorough review of our manuscript, your positive evaluation, and your constructive comments.
In accordance with your suggestions, we have substantially revised the manuscript. All revisions are highlighted in BLUE.
Comments:
- Please, clarify the PTSD as a syndrome induced by combined treatment: single pronounced stress (SPS) and laparotomy. The first sentence in the abstract is not clear; one could think that surgical trauma did not induce PTSD. In light of recent armed conflict in Ukraine, there is growing interest in the comorbidity of PTSD (Single Prolonged Stress protocol as a PTSD experimental model) and chronic low-grade inflammation, often exacerbated by additional stressors such as surgical trauma. This context underscores the importance of our study.
Specified in the text (lines 9–12).
Line 25; (after incubation) add: in pro-oxidant ascorbate-iron buffer. Specified in the text (line 25).
- Introduction: it would be worth mentioning the limitations of the use of resveratrol as a therapeutic agent (Ren Z.Q. et al. MedComm (2020). 2025 Jun 11;6(6):e70252), and the controversial results of its biological activities, especially in clinical trials (Drago L. et al. Food Frontiers. 2024;5:849–854) . Mansouri et al. cited in the manuscript [15] concluded that the dose-response meta-analysis of impact of resveratrol supplementation on sirtuin 1 level was not statistically significant. Specified in the text (lines 83–88).
- Experimental design: line126 and 127 add "body weight" (5 mg/kg body weight). Specified in the text (lines 154, 168, 169, 201, 14).
Line 116; (Group I and II) or (Control group, Group II) Corrected (line 167).
- Biochemical and Enzyme-linked immunosorbent assays.
Melatonin was listed as the study compound (that in fact were not analysed), while, interleukin-10, ceruloplasmin and cortisol were omitted and the assays were not described. Description of all the methods of biochemical determinations used in the studies, is very important. Corrected. Specified in the text (lines 210–215, 220-222).
The methods of determining of glucose, cholesterol, HDL and TG should be given in more details. ). Specified in the text (lines 223-237).
- Results:
Table 2. Are the numbers concerning glucose concentration correct because the difference between 6.47 and 4.89 seems to be significant. Corrected. Specified in the text (Table 2, lines 311-312).
Line 295: precise what incubation you mean (in pro-oxidant ascorbate-iron buffer)? Clarified in the text (lines 375-376).
Line 345: The ratio IL-10/IL-6 (not IL-6/IL-10) is usually used to assess anti-inflammatory homeostasis. It would be advisable to include this parameter in the table 1. Specified in the table 1 and the text (lines 389-392).
- Discussion: Paragraph lines 385-393. Sirt 1/AMPK and Nrf2 antioxidant pathways were not investigated in this research, so please, indicate that these information comes from literature giving an appropriate citation. Specified in the text (lines 484-488).
- Conclusions: line 409; IL-6 is pro-inflammatory cytokine, while IL-10 is anti-inflammatory cytokine (human cytokine synthesis inhibitory factor). Specified in the text (lines 511-512).
The manuscript has been substantially revised in accordance with the valuable reviewer comments.
Sincerely,
The Authors
Reviewer 2 Report
Comments and Suggestions for Authors
The manuscript describes experiments to clarify resveratrol effects on stress. To the reviewer it is unclear which emphasis the authors put onto psychological stress – judging by headline and introduction it is its main emphasis. However, for this purpose the experimental design is inappropriate. All results can easily be explained by acute surgical stress, with no need to include PTSB as an additional factor. Thus, going by the rule of the “simplest explanation possible” (Ockhams razos) the authors study resveratrol and “subchronic” somatic stress.
Human evidence for a fixed combination between PTSD and general inflammation is inconclusive. Ref. 4 states that only in a subgroup of PTSD humans chronic inflammation develops which raises the question whether in rats this link exists, does not exist, or only exist in certain animals or/and strains. The authors do not address this dilemma. In their study outline they describe a timeline of 7 days which for PTSD is too short to develop. For specific PTSD symptoms in rats (I am not aware of criteria for this phenomenon in rats but I am not an expert in psychic trauma in animals) they do not provide any evidence except a collateral assumption that this is likely.
Since the authors claim that their model is comparable to PTSD states in humans with these symptoms the discussion initially overinterprets the results. Except for the first paragraph the discussion is restricted to somatic phenomenon which are not contested but are not new either. Interstingly, in the limitations the authors themselves mention the restriction of the missing experimental group of surgical treatment only; they must also include a second treatment group with psychic stress but without surgery.
I consider the data relevant in terms of confirmation, and in this aspect they should be published. I do not agree extending the explanation to a necessary PTSD compound in the animals, without including a specific, objective criterion for the existence of PTSD. Thus, the introduction and discussion should be carefully rewritten, in order to clarify that the data support the effects of resveratrol on somatic stress parameter, and possibly may be valid for PTSD although there is no specific criterion to show this.
A few specific comments:
line 42: reference 4 only cites the link between PTSD and inflammation in a subset of people, without defining the subset.
Lines 329ff: The obtained results confirm that the coexistence of psychological stress and tissue injury triggers synergistic activation of the hypothalamic–pituitary–adrenal (HPA) axis, leading to hypercortisolemia, cytokine imbalance, and marked metabolic disturbances consistent with SIR. – overinterpreted
Ref. 27 does not specify stress sources and duration; ref. 28 is a review without original data.
Author Response
We would like to express our sincere gratitude for your thorough review of our manuscript and for your valuable comments.
In accordance with your suggestions, we have substantially revised the manuscript (revisions are highlighted in YELLOW), with particular attention to the Introduction, Materials and Methods, and Discussion sections. We have strengthened the rationale for using the Single Prolonged Stress (SPS) protocol as an experimental model of post-traumatic stress disorder (PTSD). Additionally, the Materials and Methods section now includes specific inclusion criteria based on neuroethological parameters to confirm PTSD-like status in the animals.
We acknowledge that, like most experimental models, SPS does not fully replicate the complex manifestations of PTSD in humans. However, such models remain critically important for the preclinical evaluation of therapeutic strategies.
In light of recent armed conflict in Ukraine, there is growing interest in the comorbidity of PTSD and chronic low-grade inflammation, often exacerbated by additional stressors such as surgical trauma. This context underscores the importance of our study.
As a potential therapeutic strategy, we explored the use of polyphenols, natural compounds with anti-inflammatory and antioxidant properties. One of the key challenges in their application is their poor solubility and bioavailability. In our study, we implemented a delivery method aimed at overcoming these limitations.
A few specific comments:
Line 42: reference 4 only cites the link between PTSD and inflammation in a subset of people, without defining the subset. This has now been clarified in the text (lines 44–45).
Line 329: The obtained results confirm that the coexistence of psychological stress and tissue injury triggers synergistic activation of the hypothalamic–pituitary–adrenal (HPA) axis, leading to hypercortisolemia, cytokine imbalance, and marked metabolic disturbances consistent with SIR. – overinterpreted. We have revised this sentence to reflect a more cautious interpretation (lines 425–428).
Ref. 27 does not specify stress sources and duration; ref. 28 is a review without original data. Both references have been removed.
Overall, the manuscript has been significantly revised in accordance with your valuable feedback.
Sincerely,
The Authors
Reviewer 3 Report
Comments and Suggestions for Authors
Paper titled (Resveratrol Alleviates Metabolic Disturbances and Systemic Inflammatory Response in Rats with Experimental Surgical Trauma and Post-Traumatic Stress Disorder) by Roman Ryabushko and his colleagues is an experimental study to evaluate resveratrol in a model of surgical trauma & PTSD they concluded that resveratrol alleviated the metabolic disturbances as well as systemic inflammation. In general, this is a humble trial to achieve the aim listed by the authors and I have the following recommendations for improving the msnucript
1- Title ((Resveratrol Alleviates Metabolic Disturbances and Systemic Inflammatory Response in Rats with Experimental Surgical Trauma and Post-Traumatic Stress Disorder) is clear and informative giving imprssion about the action and the mechanism
2- Abstract : I find it too long, Kindly shorten it to some extent and follow the journal guidelines
3- Introduction: is moderate in legnth but please explore the novelty and research gap for this study
4- Further clarify the aim of the study & main methods to achieve it
5- Statistics: please clarify how did you check the normality of distribution of the data & what software
6- Housing conditions and methods of reducing animal distress should be mentioned in details
7- How sample size was calculated
8- please give full references for the regimen of the drugs (dose, frequency and duration)
9- Experimnetal design is NOT clear, please clarify the groups & number of animals in each group
10- please mention the age and weight of animals
11- Methods: Concentrations of glucose, total cholesterol (CH), high-density lipoprotein (HDL), and triacyl- glycerols (TGs) were determined by enzymatic colorimetric methods using standardized reagent kits and a ULAB 101 spectrophotometer capable of measuring absorbance in the 490–600 nm range. Low-density lipoprotein (LDL) and very low-density lipoprotein (VLDL) concentrations were calculated using the Friedewald equation.
Please add the kit sources and code number
12- Mention how fasting insulin was measured
13- Table 1 please change blood serum to (serum)
14- Table 2: please write (HOMA IR index)
15- Please give a stronger conclusion and mention the limitations of your study
Please perform these changes and provide a point-to -point reply & mention the location of the changes in the manuscript
Author Response
We sincerely thank you for your thorough review of our manuscript and for your insightful comments, which have greatly contributed to improving the clarity and quality of our work.
The manuscript has been substantially revised in response to your valuable feedback (revisions are highlighted in GREEN).
1- Title ((Resveratrol Alleviates Metabolic Disturbances and Systemic Inflammatory Response in Rats with Experimental Surgical Trauma and Post-Traumatic Stress Disorder) is clear and informative giving impression about the action and the mechanism. No correction needed.
2- Abstract : I find it too long, Kindly shorten it to some extent and follow the journal guidelines. Text edited and abridged.
3- Introduction: is moderate in legnth but please explore the novelty and research gap for this study. Specified in the text (lines 109–119).
4- Further clarify the aim of the study & main methods to achieve it. Specified in the text (lines 120–122).
5- Statistics: please clarify how did you check the normality of distribution of the data & what software. Specified in the text (lines 248–255).
6- Housing conditions and methods of reducing animal distress should be mentioned in details. Specified in the text (lines 130–142, 200-201).
7- How sample size was calculated. The sample size was determined a priori for one-way ANOVA with three groups. An effect size (f = 0.4), α = 0.05, and power (1–β) = 0.80 were assumed based on previous literature and pilot data. The calculation yielded a minimum of 18 animals (6 per group). To account for possible dropouts related to surgical procedures or stress-related mortality, we included 3 additional animals, resulting in a final total of 21 rats (n = 7 per group).
8- please give full references for the regimen of the drugs (dose, frequency and duration). Specified in the text (lines 153–154, 166-170).
9- Experimental design is NOT clear, please clarify the groups & number of animals in each group. Specified in the text (lines 152–154, 166-170).
10- please mention the age and weight of animals. Specified in the text (lines 130–131).
11- Methods: Concentrations of glucose, total cholesterol (CH), high-density lipoprotein (HDL), and triacylglycerols (TGs) were determined by enzymatic colorimetric methods using standardized reagent kits and a ULAB 101 spectrophotometer capable of measuring absorbance in the 490–600 nm range. Low-density lipoprotein (LDL) and very low-density lipoprotein (VLDL) concentrations were calculated using the Friedewald equation. Please add the kit sources and code number. Specified in the text (lines 232–237).
12- Mention how fasting insulin was measured. Specified in the text (lines 226–228).
13- Table 1 please change blood serum to (serum). Completed (line 275).
14- Table 2: please write (HOMA IR index). Completed (line 275). Table 2.
15- Please give a stronger conclusion and mention the limitations of your study. Specified in the text (lines 293–500).Conclusions.
Overall, the manuscript has been significantly revised in accordance with your valuable feedback.
Sincerely,
The Authors
Round 2
Reviewer 1 Report
Comments and Suggestions for Authors
In the reviesed version of the manuscript, the authors have taken into account all my comments.
I only moticed that the ratio IL-10/IL-6 was not corrected on the pages: 26, 292 and 450.
Reviewer 2 Report
Comments and Suggestions for Authors
The authors have made many changes in the introduction and discussion to put the animal model as well as their findings into context, including a description of current uncertainties in the interplay between stress and inflammation parameters. This has considerably improved the study.
Reviewer 3 Report
Comments and Suggestions for Authors
The revised version of paper titled (Resveratrol Alleviates Metabolic Disturbances and Systemic Inflammatory Response in Rats with Experimental Surgical Trauma and Post-Traumatic Stress Disorder) by Authors Roman Ryabushko et al. was revised adequately according to this reviewer's comments
Iam glad to recommend acceptance of the current form of the manuscript